# OpenReview forum: "Phantom: General Trigger Attacks on Retrieval Augmented Language Generation"
_ICLR.cc/2025/Conference — Submitted to ICLR 2025_

### Official Review · Reviewer_SyA5 · 2024-10-30

**Soundness:** 3
**Presentation:** 3
**Contribution:** 2
**Rating:** 5
**Confidence:** 5

**Summary:**

This work focuses on the potential backdoor attacks on RAG. The authors propose an attacking method that inserts malicious data into the databases used in RAG, optimizes the data for retrieval and induces malicious behaviors. Experiments are conducted on various models and datasets including an application ChatRTX.

**Strengths:**

This work reveals an important threat that may compromise the performance of RAG. Multiple malicious behaviors are considered and evaluated. The optimization objectives are well formulated and explained. The empirical results look promising.

**Weaknesses:**

1. **Threat model**
* The threat model of local deployment and local file system is confusing and lacks practicality. Local files of the user's own are usually private and it is hard to imagine what kind of attacker can access and insert something into it. Though the authors mention something harmful like spam emails, they are usually taken care of before leveraging them in the system. The authors take for grant that malicious contents can be inserted into the database used in RAG, but I personally believe insertion itself is hard to either justify or conduct. The threat model should be more clearly and properly justified.
* When optimizing the tokens for retrieval, the embedding of retriever is assumed to be accessible to attackers, which can be a strong assumption. I would expect a more practical scenario that both retrievers and generation models are black-box and attackers can adopt some open-source models to craft examples. The totally black-box setting needs to be discussed either with empirical evaluations or explicitly discussing the implications and limitations of the current assumptions.

2. The design of s_ret and s_gen seems to be not imperceptible and is easy to detect. According to Section 4, s_ret and s_gen are sequences of tokens optimized by HotFlip and MCG respectively. Since there are no constraints to ensure the semantic meaningful, the optimized sequences can be full of non-meaningful strings. I can imagine that the perplexity score can also be much higher than ordinary content, which makes this attack easy to defend. This can be even more serious if the length of strings is getting longer. Moreover, the authors append many additional contents to the original query (s_ret, s_gen, s_cmd), the robustness of injected content can be very poor. Some case studies on the backdoored content should be provided.

3. The proportion of backdoored content in the database should be mentioned in the experiment part. I believe this proportion can be an important factor influencing both retrieval rate and final success rate. Ablation study on this proportion can help.

**Questions:**

See weakness

---

> ### Author Response · Authors · 2024-11-21
>
> We thank the reviewer for their insightful comments, and we address the main concerns  and clarifications below.
>
> **W1: Local RAG system use case**
>
> Indirect prompt injection attacks, also often referred to as XPIA (cross prompt injection attacks), are increasingly becoming more common. While we chose the use case of Chat RTX as our motivating example, this attack is not limited to local files and can be carried out against a large variety of RAG systems, as they operate on documents originating from both a user’s own file storage (either local or cloud) and Internet sources such as Wikipedia.
>
> **W1: Threat model**
>
> Our work represents one of the earliest systematic investigations into context poisoning attacks on RAG systems. Recently, however, practitioners in the field have started demonstrating the effectiveness of XPIA against major commercial RAG products [1, 2, 3], further spotlighting the relevance of this threat model.
>
> All these attacks, while simpler and less versatile than Phantom, underscore a crucial common vulnerability: Attackers can readily inject malicious content into a model's context window through various vectors. These vectors include, but are not limited to: drive-by downloads, hidden text in the body of email messages, strings encoded with ascii-smuggling [4], malicious strings embedded in HTML markup, exploitation of Markdown rendering [5], and direct social engineering. This proliferation of attack vectors validates our initial security concerns and suggests that compromising even a small portion of a model's context window is more feasible than often believed.
>
> [1] https://embracethered.com/blog/posts/2024/github-copilot-chat-prompt-injection-data-exfiltration/
>
> [2] https://embracethered.com/blog/posts/2024/google-ai-studio-data-exfiltration-now-fixed/
>
> [3] https://embracethered.com/blog/posts/2024/claude-computer-use-c2-the-zombais-are-coming/
>
> [4] https://embracethered.com/blog/posts/2024/hiding-and-finding-text-with-unicode-tags/
>
> [5] https://simonwillison.net/tags/markdown-exfiltration/
>
> **W1: White-Box Access and Transfer Attacks**
>
> We find that our attacks transfer, both to other open source RAG generators (in Appendix A.2.7) and to a Black-Box RAG system called Chat-with-RTX, developed by NVIDIA (Section 5.2). While transferability to some systems like DPR remains a challenge, the successful transfer to Chat-with-RTX shows promise, despite having no access to its retrieval component or generator model. This demonstrates that while our attacks require white-box access to a surrogate model for generating adversarial strings, they can succeed in black-box settings via direct transfer. We believe future work would focus on  improving transfer reliability across different architectures.
>
>
> **W2: Perplexity defense**
>
> The current implementation of the attack generates strings that deviate from grammatically valid English sentences, likely resulting in elevated perplexity scores when evaluated using a properly trained language model. This represents a common challenge faced by all current optimization attacks on RAG systems.
>
> While it represents a valid detection approach in the short term, perplexity-based detection introduces new complexities into the defensive landscape. Perplexity scoring requires careful calibration specific to the target content distribution, as elevated perplexity may be induced by valid content belonging to distributions unknown to the evaluation model. Moreover, it transforms the defensive problem into an optimization challenge: attackers could incorporate a perplexity minimization component into their objective functions, potentially leading to an arms race, where both detection models and attack strategies undergo continuous refinement.
>
>
> **W2 Alterations to the user’s query**
>
> Please note that our attack strategy does not alter the user’s query in any way. The adversarially crafted strings are placed only in the single corrupted document introduced in the knowledge base.
>
>
> **W3 poisoning rate**
>
> One of the core strengths of the Phantom attack is that it only requires a single poisoned passage to be injected in the knowledge base. This is negligible relative to the size of common RAG knowledge bases (for instance, MSMarco  has ~8 million passages). While the attacker can always choose to introduce more poisoned content to increase the likelihood of successful attacks, we show that this is not necessary, as the adversary can achieve high success rates with a single poisoned passage.

---

### Official Review · Reviewer_88ad · 2024-11-01

**Soundness:** 3
**Presentation:** 3
**Contribution:** 3
**Rating:** 6
**Confidence:** 5

**Summary:**

Retrieval Augmented Generation (RAG) enhances the capabilities of modern large language models (LLMs), but also provides new vectors for attackers. This paper investigates mounting a backdoor poisoning attack by injecting a single malicious document into a RAG system’s knowledge base. Specifically, the paper proposes a two-stage optimization framework to optimize the injected malicious document, with the first optimization goal being to be retrieved only when a specific trigger sequence of tokens appears in the victim’s query, and the second goal being to induce various adversarial objectives on the LLM output.

**Strengths:**

1. The study examines multiple attack objectives against RAG systems, such as Refusal to Answer, Biased opinion, Harmful Behavior, Passage Exfiltration, and validates them through extensive experiments.
2. The paper attacks a commercial black-box RAG system, NVIDIA’s Chat-with-RTX, and demonstrates that the attack achieves various objectives.
3. This paper is well-written and well-organized.

**Weaknesses:**

1. The technical contribution is not substantial enough. The paper's two-stage optimization framework is based on two existing optimization algorithms, HotFlip and GCG. Even though some improvements have been made for the attack scenario, the gap in technical contribution is not significant.
2. Lacks baselines. Although the paper proposes a new threat model of backdoor poisoning in untrusted RAG knowledge bases, straightforward baselines should be set for comparison to demonstrate the superiority of the proposed method. For example, [1], although this method is not optimized for manipulating LLM generation, it can also manipulate the subsequent behavior of the LLM by pre-defining the semantics of the retrieved malicious text.

[1] Poisoning retrieval corpora by injecting adversarial passages. EMNLP2023

**Questions:**

Please see the weaknesses for details.

---

> ### Author Response · Authors · 2024-11-21
>
> We thank the reviewer for their interesting comments. Below, we address the main concerns expressed in the review.
>
> **W1: Technical contributions**
>
> One of our main technical contributions is the design of the optimization framework for poisoning RAG systems that optimizes the retrieval and the generator attacks separately, while supporting five adversarial objectives. In addition, we made technical contributions to the design of the retrieval and generator optimization components:
>
> **Retriever contribution:** We proposed a specialized loss function that enables selective retrieval - the adversarial passage is retrieved only when a specific trigger sequence appears in the user query. While we leverage HotFlip for optimization, the novelty lies in our loss function design that precisely controls when the poisoned content is extracted by the retrieval system. The poisoned passage we design generalizes to any new queries including the trigger sequence.
>
> **Generator contribution:** We enhanced the jailbreaking capability against the LLM through our Multi-Coordinate Gradient (MCG) approach. Unlike the original GCG method, which modifies tokens sequentially, MCG optimizes multiple jailbreak tokens in parallel. This parallel optimization strategy explores the solution space more efficiently, resulting in faster convergence. We also apply the jailbreaking optimization strategy to five different adversarial objectives, unlike most jailbreaking papers that focus on generating harmful content.
>
> **W2: Baselines**
>
> **Comparison with baseline:** [1] focuses on retrieving the adversarial passage for any user query making the attack easily detectable. In our case the adversarial passage is retrieved only when a given natural trigger is present in the query, which results in a more stealthy attack. To achieve this functionality we are required to build a new loss function, which is then optimized using HotFlip.
>
> The reviewer’s suggestion, to extend the baseline [1] into  an untargeted attack for the RAG system is already addressed by concurrent work [2]. However, [2] focuses on “untargeted attacks” that are activated for any user input, whereas our method retrieves the poisoned passage only when a specific natural trigger is present in the queries, making our approach more stealthy in real-world scenarios. Secondly, their bi-level optimization approach requires more than 1000 iterations, while our two-stage optimization attack achieves high attack success in just 32 iterations - making our approach 30x more efficient.
>
> [1] Zhong et al. “Poisoning Retrieval Corpora by Injecting Adversarial Passages”, EMNLP’23Zhong et al. “Poisoning Retrieval Corpora by Injecting Adversarial Passages”, EMNLP’23
>
> [2] Tan et al. "Glue pizza and eat rocks"--Exploiting Vulnerabilities in Retrieval-Augmented Generative Models." The 2024 Conference on Empirical Methods in Natural Language Processing, arXiv’24.

---

> > ### Comment · Reviewer_88ad · 2024-11-26
> > **Official Comment by Reviewer 88ad**
> >
> > Thank you for your response. I will maintain my positive score.

---

> > > ### Author Response · Authors · 2024-11-26
> > >
> > > Thank you for your response! We have added detailed comparisons with three concurrent works in Appendix A.1 of the revised version of the paper, which also includes the Tan et al.'24 work that we discussed above.

---

### Official Review · Reviewer_aMuq · 2024-11-04

**Soundness:** 2
**Presentation:** 2
**Contribution:** 2
**Rating:** 3
**Confidence:** 4

**Summary:**

This paper introduces Phantom, a framework that proposes a novel attack on Retrieval-Augmented Generation (RAG) systems by injecting malicious documents into their knowledge bases, which are then retrieved by specific trigger sequences to manipulate outputs. The attack utilizes a two-stage optimization process: firstly, crafting a document that aligns with the RAG's retrieval mechanics when triggered, and secondly, generating text to induce specific adversarial outcomes like misinformation or refusal to answer. Extensive experiments demonstrate the effectiveness of Phantom across various large language models and datasets, revealing vulnerabilities in both proprietary and open-source RAG implementations. The paper also discusses potential mitigations, stressing the challenge of balancing system security with the utility of RAG systems in real-world applications.

**Strengths:**

- The experiments of this paper are extensive.
- The security of the RAG system is an important topic.

**Weaknesses:**

1. The threat model lacks clarity, particularly regarding the attacker's capabilities. Line 153 suggests the attacker does not need knowledge of other documents to initiate the attack. However, the Multi-Coordinate Gradient (MCG) technique described on lines 274-275 requires access to the top-k passages retrieved in response to m user queries. These passages, presumably from the RAG databases, are crucial for optimization. Appendix A.3.5 explores transferability between different datasets, implying the use of a consistent dataset (MS MARCO) for both optimization and attack phases. This suggests that the attacker must have knowledge of the RAG database to effectively deploy the MCG technique.

2. The paper lacks a comprehensive comparison with prior works. Generating malicious passages retrievable exclusively by queries with specific triggers has been explored by [1], and inducing target outputs in LLMs using suffix generation has been studied by [2]. This paper does not compare its methods against [1] concerning Retrieval Failure Rate, nor does it compare attack success rates with [2]. Additionally, the authors should consider citing [2], which introduces a bi-level optimization for generation and retrieval conditions and employs gradient-based optimization to concatenate strings for retrieval and generation.

3. The novelty of the second method (MCG) appears limited. It is primarily a variant of the GCG with modifications only detailed in lines 7-8. Although MCG reportedly outperforms GCG, the reasons for this improvement and the rationale behind this incremental change are not well-explained.

4. The loss modeling for MCG is problematic. Equation 2 assumes that only one passage is retrieved and used as input for the generator. However, typical RAG applications retrieve multiple passages for context. Notably, Appendix A.2.26's Table 9 shows that the position of $S_{gen}$ significantly affects its efficacy, with prefixes performing better than suffixes. In scenarios where multiple passages are retrieved, there's no guarantee the malicious passage will be at the top-1 position, undermining the experiment's validity if $S_{gen}$ is not the prefix of the entire context. Why the string $S_{gen}$ optimized on a single passage could be effective in multiple passage settings is not clear to me

5. There is a noticeable absence of experiments involving other retrievers such as DPR, BGE, REALM, or ORQA. While the paper discusses the retrieval failure rate on DPR in the appendix, it does not present end-to-end attack success rates for these retrievers in RAG systems. The high retrieval failure rates on other retrievers besides Contriever suggest limited practicality and effectiveness of the attacks.

6. Could the authors provide explanations for the high retrieval failure rates observed with DPR and Contriever-MS? It is particularly puzzling for Contriever-MS, a version of Contriever fine-tuned on the MS MARCO dataset, to show a significant performance discrepancy compared to its base model.

7. The paper discusses an attack against the Chat RTX without detailed knowledge of its retriever. Given the poor transferability between retrievers noted in lines 1105-1107, the successful attack using Contriever on the black-box Chat RTX system is perplexing. Could the authors provide the specific success rates achieved against it?

[1] Xue, Jiaqi, et al. "BadRAG: Identifying Vulnerabilities in Retrieval Augmented Generation of Large Language Models." arXiv preprint arXiv:2406.00083 (2024).

[2] Tan, Zhen, et al. "" Glue pizza and eat rocks"--Exploiting Vulnerabilities in Retrieval-Augmented Generative Models." The 2024 Conference on Empirical Methods in Natural Language Processing (EMNLP 2024)

**Questions:**

Please refer to the weakness part.

---

> ### Author Response · Authors · 2024-11-21
> **Clarifications on design of Phantom and comparison with other works (1/2)**
>
> We thank the reviewer for their comments. The following paragraphs address the main comments and provide clarifications on the design of Phantom.
>
> **W1: Attacker’s knowledge.**
>
> Thank you for pointing out the assumption. While passages are sampled from MS-Marco, MCG optimization is conducted on OUT queries (unrelated to the target trigger) with the trigger artificially appended. This ensures that the top-$k$ retrieved passages ($k=5$) are unrelated to the trigger sequence, supporting the assumption that no prior knowledge of trigger-related documents is required.
> To further validate that our attack works without relying on passages from the dataset, we fill the top-k passages for MCG optimization with synthetically generated passages by an external oracle, such as GPT-4, relevant to the query. Testing on two triggers, "xbox" and "lebron james," we observe similar effectiveness as our original experiments, with attack success rates of 88% and 79%, respectively.
>
> **W2: Concurrent work.**
>
> While we appreciate the comparison with these works, they should be considered concurrent given the timeline of these submissions. We address the comparisons separately:
>
> **Regarding [1]:** Our attack demonstrates broader applicability, successfully targeting 5 different adversarial objectives compared to only two in [1]. A crucial distinction is that our attack remains effective even when the adversarial objective is in conflict with the RAG generator's safety alignment (e.g., "Threaten the user"). In contrast, their approach does not work in such scenarios-due to the attacker's inability to circumvent the generator’s safety alignment.
> Additionally, their attack relies on unnatural triggers for context leakage and tool usage attacks - likely due to jailbreaking constraints, whereas we don't have such limitations. While both works formulate the retriever loss function in a similar fashion, our end-to-end attack overcomes the key limitations discussed above. Finally, we require only one poisoned passage to be added into the RAG database while [1] requires around 10 poisoned passages for their attack to succeed.
>
> **Regarding [2]:** There are fundamental differences in both approach and efficiency:
>
> Attack Scope: Their work focuses on untargeted attacks that are activated for any user input, whereas our method activates only when a specific natural trigger is present in the queries, making our approach much more stealthy in real-world scenarios.
>
> Optimization Strategy: They propose a bi-level optimization that alternates between HotFlip (for retriever tokens) and GCG (for generator tokens) and requires over 1000 iterations for the attack to succeed. They don't provide any empirical justification for why their approach is superior to a sequential two-step optimization. Our two-step attack, on the other hand, achieves high attack success in just 32 iterations - making our approach 30x more efficient than [2].
>
> Empirical Findings: Their work also observed similar issues when attacking Contriever-MS Marco compared to base Contriever, which supports our findings about retrievers having different robustness to poisoning.
>
> We have added detailed comparisons with three concurrent works in Appendix A.1 of the revised version of the paper, which also includes the two works that we discussed above.
>
> **W3: Rationale for MCG.**
>
> The Multi-Coordinate Gradient (MCG) approach, while built upon GCG, introduces a strategic modification to address the limitation of token-by-token optimization. Our primary intuition behind MCG was to enable more efficient exploration of the optimization space by modifying multiple tokens simultaneously in early iterations, rather than being constrained to single-token changes as in GCG. As the optimization progresses, MCG naturally transitions to modifying fewer tokens, eventually converging to GCG-like behavior. This allows us to converge faster with fewer iterations, making our attack more efficient to run. (Table 7, Appendix A.2.4)

---

> ### Author Response · Authors · 2024-11-24
> **Clarifications on design of Phantom and comparison with other works (2/2)**
>
> **W4: Loss modeling.**
>
> We would like to clarify that Equation (2) does not assume that only one passage is retrieved.  Our approach is explicitly designed and tested for multi-passage retrieval scenarios, with all experiments consistently using k=5 for retrieved passages. During MCG optimization, each query has a corresponding context which consists of 4 passages relevant to the query content plus one adversarial passage, all ranked based on their similarity scores. Additionally, the prefix/suffix positioning discussed in Appendix A.2.26 (Table 9) refers specifically to token arrangement within the adversarial passage itself - meaning the arrangement of (sret+sgen+adv_cmd) vs (sret+adv_cmd+sgen) is internal to the malicious passage and independent of where the adversarial passage appears in the overall retrieved context. This clarification should resolve the confusion, as our method is explicitly designed and validated in a realistic multi-passage retrieval setting, not just the simplified single-passage case.
>
> **W5: Retrievers.**
>
> We evaluate our attacks on retrievers commonly studied in both published and concurrent work [1,2,3,4]. These works have also observed lower success rates for both targeted and untargeted attacks on retrievers such as Contriever-MS and DPR pointing to varied robustness of retrievers against adversarial attacks. Also note that, we add only a single poisoned passage unlike many works [1,2,3] that require multiple poisoned passages, and still achieve  substantial retrieval rate exceeding 50% on these retrievers. One could further boost the attack success similar to prior work by adding more poisoned passages such that at least one of adversarial passages shows up in the top-k, improving the reported metrics. The goal of our work was to minimize poisoning while having an effective attack.
>
> **W6: Contriever-MS.**
>
> The high retrieval failure rates observed with Contriever-MS, particularly the significant performance gap between Contriever-MS and its base model, align with findings from concurrent works [1],  [2] and [3]. These observations suggest that fine-tuning on structured datasets like MS MARCO might introduce additional robustness against adversarial attacks. Understanding the impact on robustness of retrievers via fine-tuning with structured dataset, would be an interesting topic of future work.
>
> **W7 chat with RTX.**
>
> We successfully attack the black-box NVIDIA Chat-with-RTX system. While transferring between retrievers is challenging, our attack transferred from Contriever to Chat-with-RTX. We don’t have details on the retriever used by Chat-with-RTX, but we suspect it uses a similar architecture to Contriever.
>
> We ran additional experiments on ChatRTX using the trigger word “confidential”. This trigger word is especially fitting because it is in the top 0.05% most frequent words in the Enron dataset (~500k documents), used as the knowledge base, appearing a total of 24.5k times. Thus, the relatively high frequency of natural occurrences increases the difficulty of successfully inducing the retriever to select the corrupted passage.
>
> To test Phantom on the Mistral 7B int4 model in ChatRTX, we used 25 random queries from the MSMARCO dataset that contain the trigger word. Of the 25 queries, the adversarial passage was retrieved as the top-1 document from the knowledge base 15 times, yielding a retriever failure rate of 40% in a true black-box setting. Of the 15 queries where the adversarial passage was retrieved, the adversarial objective (Passage Exfiltration) was executed 12 times, leaking the model’s context, for a total attack success rate of 48%.
>
> [1] Xue, Jiaqi, et al. "BadRAG: Identifying Vulnerabilities in Retrieval Augmented Generation of Large Language Models." arXiv preprint arXiv:2406.00083 (2024).
>
> [2] Tan, Zhen, et al. "" Glue pizza and eat rocks"--Exploiting Vulnerabilities in Retrieval-Augmented Generative Models." The 2024 Conference on Empirical Methods in Natural Language Processing (EMNLP 2024)
>
> [3] Zou et al. “PoisonedRAG: Knowledge Corruption Attacks to Retrieval-Augmented Generation of Large Language Models” (USENIX’25)
>
> [4] Zhong et al. “Poisoning Retrieval Corpora by Injecting Adversarial Passages”, EMNLP’23

---

### Meta-Review · Area_Chair_tja5 · 2024-12-14

**Metareview:**

The paper shows an adversarial threat against the retrieval augmented generation (RAG) system. Specifically, it constructs a backdoor in a document, and the language model would not generate the correct response when the trigger appears. The paper designs a two-stage optimization framework that both improves GCG attack's efficiency and keeps the normal functionality where the trigger is not shown. While the task is very important and the experiments show promising results, the paper still has several major concerns solved before publication. First, the threat model must be clarified further, especially in real-world applications. The current threat model shares a similarity with the jailbreak attack so it makes the paper's novelty and contributions less significant. Second, I encourage the author to include some related works in the discussion as discussed in the rebuttal, which can further clarify the novelty of the proposed methods. Third, it will make the paper better if the author can include some discussions with potential defenses and include more baselines with different retriever and attack methods.

**Additional Comments On Reviewer Discussion:**

All reviewers show concerns about the novelty and insufficient experiments. Reviewer aMuq's concerns are mostly in previous works and some experimental settings. Reviewer SyA5 raises concerns over the threat model. I greatly appreciate the effort the authors made to address the concerns raised by reviewers. Some concerns about the experiments are solved however I believe the novelty and threat model concerns are still valid and they also make the paper's novelty and contributions less significant. As suggested by the metareview, I believe the paper can be much improved by addressing these concerns for future publication.

---

### Decision · Program_Chairs · 2025-01-22

Reject